

# Modelling feedbacks between human and natural processes in the land system

*Derek T. Robinson[1], Alan Di Vittorio[2], Peter Alexander[3,4], Almut Arneth[5], C. Michael Barton[6], Daniel G. Brown[7], Albert Kettner[8], Carsten Lemmen[9], Brian C. O'Neill[10], Marco Janssen[11], Thomas A. M. Pugh[12], Sam S. Rabin[5], Mark Rounsevell[3,5], James P. Syvitski[13], Isaac Ullah[14], Peter H. Verburg[15]

[1]Department of Geography and Environmental Management, University of Waterloo, Waterloo, Ontario, Canada.

[2]Climate and Environmental Sciences Department, Lawrence Berkley National Laboratory, Berkeley, California, USA.

[3]School of Geosciences, University of Edinburgh, Drummond Street, Edinburgh EH8 9XP, United Kingdom.

[4]Land Economy and Environment Research, SRUC, West Mains Road, Edinburgh EH9 3JG, United Kingdom.

[5]Karlsruhe Institute of Technology, Institute of Meteorology and Climate Research – Atmospheric Environmental Research (IMK-IFU), Garmisch-Partenkirchen, Germany.

[6]School of Human Evolution & Social Change, Arizona State University, Tempe, Arizona, USA.

[7]School of Natural Resources and Environment, University of Michigan, Ann Arbor, Michigan, USA.

[8]Dartmouth Flood Observatory, [12]Community Surface Dynamics Modeling System, Institute of Arctic and Alpine Research, University of Colorado, Boulder, Colorado, USA.

[9]Institute of Coastal Research, Helmholtz-Zentrum Geesthacht, Geesthacht, Germany.

[10]National Center for Atmospheric Research, Boulder, Colorado, USA.

[11]School of Sustainability, Arizona State University, Tempe, Arizona, USA.

[12]School of Geography, Earth and Environmental Sciences, University of Birmingham, Edgbaston, Birmingham, United Kingdom.

[13]Community Surface Dynamics Modeling System, University of Colorado, Boulder, Colorado, USA.

[14]Department of Anthropology, San Diego State University, San Diego, California, USA.

[15]Environmental Geography Group, Institute for Environmental Studies, VU University Amsterdam, Amsterdam, The Netherlands.

*Corresponding author email: dtrobins@uwaterloo.ca



## ABSTRACT

The unprecedented use of Earth's resources by humans, in combination with the increasing natural variability in natural processes over the past century, is affecting evolution of the Earth system. To better

understand natural processes and their potential future trajectories requires improved integration with and quantification of human processes. Similarly, to mitigate risk and facilitate socio-economic development requires a better understanding of how the natural system (e.g., climate variability and change, extreme weather events, and processes affecting soil fertility) affects human processes. To capture and formalize our understanding of the interactions and feedback between human and natural

systems a variety of modelling approaches are used. While integrated assessment models are widely recognized as supporting this goal and integrating representations of the human and natural system for global applications, an increasing diversity of models and corresponding research have focused on coupling models specializing in specific human (e.g., decision-making) or natural (e.g., erosion) processes at multiple scales. Domain experts develop these specialized models with a greater degree of

detail, accuracy, and transparency, with many adopting open-science norms that use new technology for model sharing, coupling, and high performance computing. We highlight examples of four different approaches used to couple representations of the human and natural system, which vary in the processes represented and in the scale of their application. The examples illustrate how groups of researchers have attempted to overcome the lack of suitable frameworks for coupling human and natural systems to

answer questions specific to feedbacks between human and natural systems. We draw from these examples broader lessons about system and model coupling and discuss the challenges associated with maintaining consistency across models and representing feedback between human and natural systems in coupled models.



# 1. INTRODUCTION

Models designed to improve our understanding of human-environment interactions simulate interdependent processes that link human activities and natural processes, but usually with a focus on the

human or natural system. When simulating the land system, such models tend to incorporate either detailed decision-making algorithms with simplified ecosystem responses (e.g., land use models) or simple mechanisms to drive land-cover patterns that affect detailed environmental processes (e.g., ecosystem models). These one-sided approaches are prone to generating biased results, which can be improved by capturing the feedbacks between human and natural processes (Verburg 2006, Evans et al.,

2013; Rounsevell et al., 2014). Hence, improving our understanding of the interdependent dynamics of natural systems and land change through modelling remains a key opportunity and important challenge for Earth systems research (NRC 2013).

     Land use describes how humans use the land and the activities that take place at a location (e.g., agricultural or forest production), whereas land-cover change describes the transition of the physical

surface cover (e.g., crop or forest cover) at a location. These distinct concepts are inextricably linked, and models sometimes conflate them or, when represented separately, fail to link them. Because of the tradition of division between human and natural sciences (Liu et al., 2007), land change science and social science have focused on how socioeconomic drivers interact with environmental variability to affect new quantities and patterns of land use (Turner II et al. 2007) while natural science has focused on

modelling natural-system responses to prescribed land-cover changes (e.g., Lawrence et al., 2012).





An important limitation of most natural system models[1] is that the impacts of human action are represented through changes in land cover that rarely involve mechanistic descriptions of the human decision processes driving them. These models are typically applied at coarse resolutions and ignore the influence of critical land management activities on natural processes and micro-to-regional climate

5      associated with fine-resolution factors such as landscape configuration (e.g., Running and Hunt 1993; Smith et al. 2001, 2014; Robinson et al. 2009), fragmentation and edge effects (e.g., Parton et al. 1987; Lawrence et al. 2011), and horizontal energy transfers (e.g., Coops and Waring 2001). The consequences of excluding these factors on the representation of natural processes can be significant because they aggregate to affect global processes.

10      Conversely, efforts to model and represent changes in how land is used by humans (i.e., land-use change models, LUCMs) have been developed to understand how human processes impact the environment, but in ways that often over-simplify the representation of natural processes (Evans et al. 2013). While such models vary in their level of process detail, they usually include some representation of the economic and social trade-offs associated with alternative land-use types. Over the past 5-10

15      years, the representation of natural systems has been improved in LUCMs by systematically increasing the complexity of natural processes represented from inventory approaches to rule-based approaches (e.g., Manson 2005), statistical models (e.g., Deadman et al. 2004), dynamic linking to ecosystem models (e.g., Matthews 2006, Yadav et al. 2008, Luus et al. 2013), or coupling of integrated assessment

---

[1] Natural system model is used as an overarching term for Earth system, land surface, ecosystem, and more specific models of natural processes (e.g., erosion). We use the following nomenclature: a) Earth System models couple land and ocean biogeochemistry to atmospheric processes, and represent surface-atmosphere interactions, such that $CO_2$ respiration (and other processes) affects the atmospheric $CO_2$ concentration, which in turn affects vegetation growth; b) ecosystem models integrate biogeochemistry, biophysical processes (e.g., latent and sensible heat fluxes), and vegetation structure to simulate dynamic terrestrial vegetation growth (Kucharik et al. 2000); and c) land surface models represent heat and moisture fluxes between the land surface and atmosphere and can include vegetation properties using anything from simple parameters (e.g., Bonan 1996) to detailed ecosystem models.



models and Earth-system models (e.g., Collins et al., 2015). Even with the impetus to better understand human-environment interaction through model coupling, land-use science and the natural sciences remain disparate fields (Liu et al. 2007). Novel integrative modelling methods are being developed to create technical frameworks for and intersecting applications between these two communities (e.g., Hill

et al. 2004, Lemmen et al. 2017, Peckham et al. 2013, Robinson et al. 2013, Collins et al. 2015, Barton et al. 2016) that offer insight and an initial benchmark for identifying methods for improvement.

The promise of greater integration between our representations of land-use and natural (land cover) processes, and therefore between social and natural system dynamics, lies partly in the spatially distributed representations of land use, land cover, vegetation, climate, and hydrologic features. Models

in land-change and natural sciences tend to contain a description of the land surface (often gridded) and, while the representations of these systems may differ in their level of detail, they are often complementary, thus facilitating a more complete representation and understanding of land-surface change through integration. The coupling of land-change and natural-system models promises a new approach to characterizing and understanding humans as a driving force for Earth-system processes

through the linked understanding of land use and land cover as an integrated land system.

The potential gains from greater coupling are threefold. First, the unprecedented use of Earth's resources by humans, which is considered unsustainable in many respects, is affecting evolution of the Earth system (Zalasiewicz et al. 2015, Waters et al. 2016). Potential futures of the natural environment need to account better for human processes (Bai et al. 2015). Second, the natural system (e.g., climate

variability and change, extreme weather events, processes affecting soil fertility) also affects human processes. Therefore, interactions and feedback within the social and in socio-ecological systems must be better quantified (Verburg et al. 2016). Third, to achieve these substantive gains requires an understanding and critical assessment of the different modelling approaches to coupling representations

of human systems with natural systems that span from local ecological and biophysical processes (e.g.,

erosion, hydrology, vegetation growth) through to global processes (e.g., climate). Through model

development and integration, knowledge is synthesized, formalized in mathematics or computer code,

communicated and discussed among others, and gaps in knowledge and data are identified that create an

iterative modelling process (Rounsevell et al. 2012). Finally, coupled models will be most useful if we

can use them to test possible interventions in the Earth system that might help avoid the worst possible

outcomes.

Coupled modelling contrasts with integrated modelling (see Verburg et al. 2016), which combines

simplified versions of both the natural and human system and explicitly incorporates feedbacks between

the two systems. It might be that coupling models leads to different conclusions, and this should be

tested. In contrast to integrated modelling, coupled modelling combines specialized models instead of

relying on simplifications of the specialized models within an integrated framework. Furthermore,

within the new norms of open science, new technology for model sharing, model coupling, and high

performance computing make it possible to connect specialized models, which was not possible when

Integrated Assessment Models (IAMs) were first conceptualized 25 years ago. The coupling approach

enables a greater degree of transparency and accuracy in coupled models.

We present multiple approaches to coupling land-change and natural-system models to evaluate how

alternative approaches to representing feedbacks add value to scientific inquiry into global change and

thus generate new insights into the sustainable management of human-environment interactions. Based

on the current state of the science, we categorize conceptual approaches to coupling land-change models

with natural-system models that differ across a range of spatial extents and coupling methods. Using

four case studies, we critically assess the influence of land-change processes on natural-system

processes and vice versa, focusing on the implications of these feedbacks for system dynamics, the



research questions that model coupling enables, and the strengths and weaknesses of the coupling approach. After presenting the case studies of coupled land-change and natural-system models, we describe the lessons learned from the various approaches, the different types of consistency that should be maintained between coupled models as well as the feedbacks represented between the human and

natural systems.

## 1.1 Technical approaches to coupling Earth-system and land-change models

When two models communicate in a coordinated fashion, they form a coupled model, where the constituents are often termed components (Dunlap 2013, Dunlap et al. 2013). One of the first examples

of coupled models occurred in the 1970s to describe the interaction of different physical processes represented by numerical models for weather prediction (e.g. Schneider and Dickinson 1974). This initial coupling of processes is still in use today, e.g. in the Modular Earth Submodel System (MESSy, Jöckel et al. 2005). More often, however, the term is used for domain coupling, i.e., the coordinated interaction of models for different Earth-system domains or "spheres." A typical ecological component

would describe the ecosystems and biogeochemistry of the biosphere in a coupled model that also includes representations of the atmosphere, hydrosphere and cryosphere, thus constituting a so-called Earth-system model.

Model coupling can be described by the strength and frequency of interaction between two software components, often placed in a continuum between 'tight' and 'loose' where tight is a coupling with high

coordination and frequent communication and loose has low coordination and rare communication. While the simple categorization between tight and loose neglects the multidimensionality of tightness and looseness (e.g., along different axes of coordination, communication, code integration), it is sufficient to address the current state of the art for coupling land-change and natural-system models.



In a loose coupling, communication is mostly based on the exchange of data files, and coordination is the automated or manual arrangement of independently operating (and different) components and externally organized data exchange. No interaction of the developers of the components is required, and coupling can extend across different expert communities and platforms. In a tight coupling - some call

this the monolithic approach - all components exist within a single model and are interdependent; they share much of their programming code and access shared memory for communication, all coordination is programmed into a monolithic model.

Many of the current coupled models have intermediate degrees of coupling, and the different existing technologies attempt to utilize the strengths of tight and loose coupling approaches in different

ways (Syvitski et al. 2013), often with compromises between control versus openness, high performance computing versus wide distribution, distributed versus concentrated expertise, or shared versus modular independent code. Coupling frameworks are typically introduced when the simulation environment becomes multidisciplinary and requires collaborative modeling of several scientific disciplines, which would be too complex to be comprehended by a single individual or research group (Voinov et al.,

2010). For example, the Community Surface Dynamics Modeling System (CSDMS, Peckham et al. 2013) promotes distributed expertise and independent models in the domain of Earth-surface dynamics. In it, all components are required to implement basic model interfaces (BMI) as communication ports with any other components in CSDMS (Syvitski et al. 2014). The Modular System for Shelves and Coasts (MOSSCO, Lemmen et al. 2017 also promotes distributed expertise and independent models;

similarly it is suggested that all components implement the Earth System Modeling Framework (ESMF, Hill et al. 2004) protocol, but the focus is on immediate deployment on high performance computing infrastructure.

Multidisciplinary modelling and model coupling can be improved when a common language is used. For example, traditional coupling between ocean and atmosphere in Earth System Models typically use the Climate and Forecast conventions (Eaton et al. 2011). A controlled vocabulary in these conventions assists understanding of model processes and facilitates their coupling among models or replacement in new models. With a similar goal but different approach, CSDMS introduced rules for creation of unequivocal terms through their standard names system that functions as a semantic matching mechanism for determining whether two terms refer to the same quantity with associated predefined units. This concept is currently undergoing transition to a Geoscience Standard Names ontology that reaches out to include social science terms (David et al. 2016), which can benefit communication between communities (i.e., natural and social science) that may have different terms and descriptions of similar processes (Di Vittorio et al. 2014).

With a common language, data can be more easily and unambiguously communicated between components in a coupled system. As in any communication, the reciprocal exchange of data, i.e. a feedback, may serve to amplify, dampen, or constrain the behaviour of modelled processes and their outputs. Understanding and controlling the effects of these feedbacks is a challenge in every coupled model system and a design choice that is visible in the feedback structure employed in the specific coupling strategy.

## 2. Examples of approaches to coupling

We showcase four examples of different approaches used to couple representations of the human and natural system, which vary in the processes represented as well as in the scale of their application. The



examples illustrate how groups of researchers have attempted to overcome the lack of suitable frameworks for coupling human and natural systems.

### 2.1 Effects of subsistence agriculture and pastoralism on erosion

2.1.1 Definition and description

The Mediterranean Landscape Dynamics (MedLand) project has developed a computational laboratory for high-resolution modeling of land-use/landscape interaction dynamics in Mediterranean landscapes. The MedLand Modeling Laboratory (MML) is designed as a controlled experimental environment in

which to model coupled human and natural systems (Bankes et al., 2002; Miller and Page, 2007; van der Leeuw, 2004; Verburg et al., in press). The MML integrates an agent-based model (ABM) of households practicing subsistence agriculture and/or pastoralism and a cellular automata landscape evolution model (LEM) that simulates the dynamics of surface erosion/deposition, vegetation change, and soil depth/fertility at the scale of local watersheds. Observational or modeled climatic data are used as input

to the LEM (Barton et al., 2010, 2012; Mayer and Sarjoughian, 2009; Mitasova et al., 2013; Soto-Berelov, 2011). The components of the MML are modular and are dynamically coupled through an interaction model to manage overall model complexity (Davis and Anderson, 2004, p. 200; Gholami et al., 2014; Sarjoughian et al., 2013; Sarjoughian, 2006; Sarjoughian et al., 2015) (See S1 1.1).

Villages and household actors are represented as agents in an ABM that simulates land-use decisions

and practices, mirroring the organization of known small-scale subsistence farmers (Banning, 2010; Flannery, 1993; Kohler and van der Leeuw, 2007). These agents select land for cultivation and grazing, using decision algorithms and projected returns informed by studies of subsistence farming, with emphasis on the Mediterranean and xeric landscapes (Ullah 2017).

The LEM iteratively evolves digital terrain, soil, and vegetation on landscapes by simulating sediment entrainment, transport, and deposition using a 3D implementation of the Unit Stream Power Erosion/Deposition (USPED) equation and the Stream Power equation (Barton et al., 2016; Mitasova et al. 2013). The LEM also tracks changes in soil depth and fertility due to cultivation and fallowing. A

simple vegetation model simulates clearance for cultivation or removal by grazing and regrowth tuned to a Mediterranean 50-year succession interval based on empirical studies in the region (Bonet, 2004; Bonet and Pausas, 2007).

2.1.2 Feedbacks

The MML provides for three general types of model coupling, which differ in the amount and frequency of feedback between sub-models during a simulation (Figure 1). The simplest and first type of coupling is very loose and indirect (Section 1.1), whereby a predefined land-use scenario is "injected" into a coupled model of land-cover and landscape evolution at the start of the simulation. There are no feedbacks between land-use and land-cover change (nor with any other natural processes; Figure 1-a).

The static nature of the land cover results in simplistic linear growth of erosion and deposition (ED) over time and variability in ED is observed through the implementation of different land-use scenarios.

The second type of coupling simulates a sequence of prescribed land-use and land-cover change events resulting from human decision algorithms. The ABM generates a new land-use scenario at each time-step, which is "injected" into the LEM as it proceeds, but environmental repercussions of land-

cover change do not affect land-use decisions. This uni-directional coupling is relatively deterministic and capable of producing non-linear ED (Figure 1-b). Because the coupling is one-way, the human and natural systems models may be run independently and the land-use data used as an input into the LEM at a later time.

A final coupling mode available in the MML enables bidirectional feedback. In this mode, the ABM continually polls the state of the land cover and biophysical changes (i.e., erosion) from the LEM and uses that information to update the land-use configuration (Figure 1-c). As with the uni-directional coupling strategy, human land-use is dynamic and inserted back into the LEM at each time-step.

However, in this method of coupling, landscape changes generated in the LEM influence human decision-making. The incorporation of feedback not only produces non-linearity in simulation output, but it also allows for the emergence of significant inter-run variation that more accurately captures the range of potential outcomes from the initial starting scenario.

The flexibility of coupling strategies within the MML allows for varying degrees of information

exchange to be simulated between human and natural-system processes. This allows a wide range of simulation capabilities. Simpler couplings allow faster (or longer, or more) simulations at the expense of induced linearity in model results. More complex couplings allow for non-linearity and/or an increased understanding of variability, but require longer run-times and tighter integration of model sub-systems, potentially reducing the modularity of the modeling environment. In practical terms, this allows the

simulation complexity to be increased or decreased in response to specific research questions.

### 2.1.3 Case study

We carried out four experiments (Table 1), simulating 200 years of landscape dynamics around a simulated Neolithic farming village in Southern Spain (See SI 1.2). The experiments evaluated the

consequences of coupling human and natural systems using four approaches based on the above coupling effects on soil ED. These were: 1) no human presence, a control model of natural vegetation; 2) human system creates static land-use scenarios, which are injected into the natural-system model at start-up; 3) uni-directional, human impacts on natural-system model at each step of the simulation, and





4) bi-directional feedbacks at each step of the simulation, in which humans alter the biophysical system and those biophysical changes affect human actions. For simplicity and clarity, these simulations were conducted with non-changing climate conditions. Following procedures used in prior research with the MML, we measured landscape dynamics in these experiments by calculating the amount of sediment deposited and amount eroded from grid cells in the model landscape.

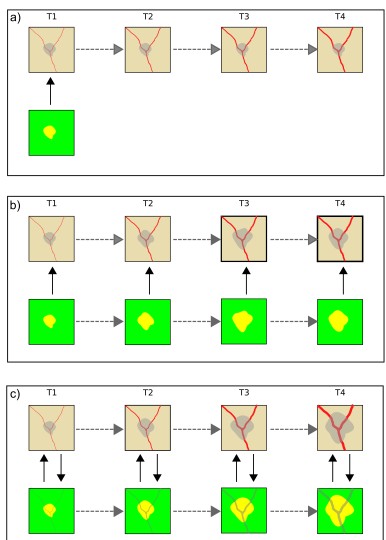

Figure 1: Schematic diagram of the three types of model coupling. (a) Loose coupling and no feedback, as used in Experiment 2 (Table 1). A static land-cover configuration is inserted into the LEM at the start of the simulation and is never changed. (b) Tighter coupling and uni-directional communication as used in Experiment 3 (Table 1). Human land-use is dynamic, creating a new land-cover configuration that is inserted into the LEM at each time-step. (c) Tighter coupling and bidirectional feedbacks, as used in Experiment 4 (Table 1). Dynamic human land-use is now also effected at each time-step by landscape changes, introducing wider variability in the dynamics of both human land-use and landscape evolution.



Table 1: Experiments conducted showing the type of human land-use simulated, the type of coupling between the human land-use model and the landscape evolution model, and the strength of that coupling.

| Experiment | Land-Use Type | Coupling Type | Coupling Strength |
|---|---|---|---|
| 1) Natural land-cover | None | No coupling | None |
| 2) Static land-use | Fixed configurations | Input parameter | Extremely weak |
| 3) Static population | Untenured, randomized yearly | uni-directional | Weak |
| 4) Dynamic population | Tenured, maximized yearly | bi-directional | Strong |

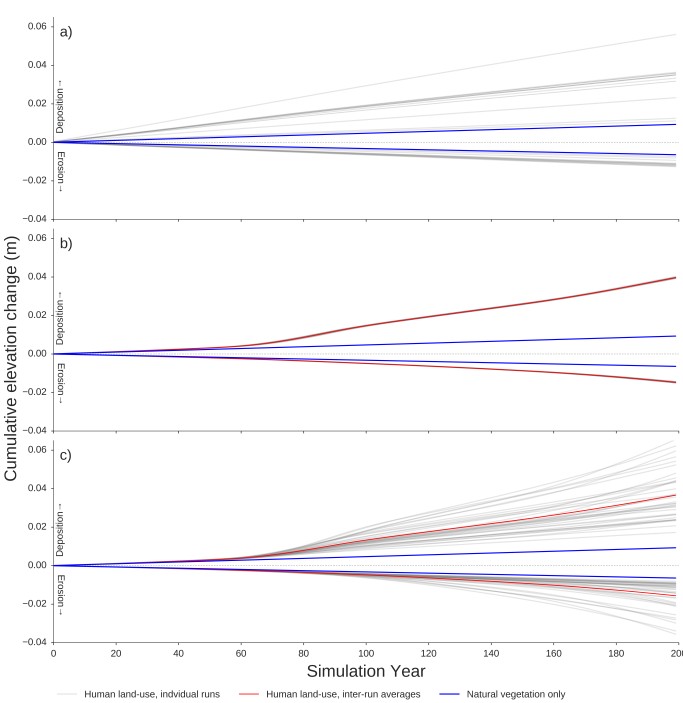

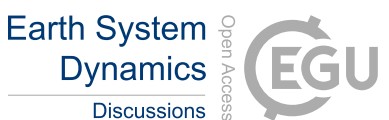
Figure 2: Mean cumulative erosion and deposition (ED) per landscape grid cell for each experiment. Values above 0.0 are cumulative mean deposition per grid cell across the watershed. Values below 0.0 are cumulative mean erosion per grid cell across the watershed. The thin grey lines represent the

temporal trends in ED for each run of each individual experiment (i.e., the four experiments in Table 1). The solid black lines represent the average temporal trends across all runs of experiments that had multiple runs. The dashed black lines represent the "baseline" experiment (experiment 1 in Table 1) that used only naturally wooded vegetation (i.e., no human land-use). (a) Results for fixed configurations of human land-use that do not change over time (experiment 2 from Table 1). These result in linear

temporal trends in ED over time. (b) Results from the untenured, randomized style of human land-use with a fixed human population target unaffected by landscape change (experiment 3 from Table 1). These produce non-linear temporal trends, but very little inter-run variability. (c) Results from the tenured, maximizing style of human land-use with a dynamic population affected by landscape evolution (experiment 4 from Table 1). These also produce non-linear temporal dynamics in ED, and also generate

a wide range of inter-run variability

        The results of the land-use simulation with a loose, indirect coupling scenario (Experiment 2 in Table 1) are shown in Figure 2-a. This consisted of 10 simulation runs of distinct configurations of input land-cover representing increasing human pressure in the farming and grazing catchment, but which remained

static for the entire 200 years. Without feedback between the human and natural-system model, each configuration results in a single, linear temporal trajectory in accumulated ED over the 200 years of the simulation. The most intensely altered land-cover configurations resulted in the largest cumulative

amounts of ED, and all of the human-altered land-covers resulted in larger ED impacts than in Experiment 1, which contained static forest cover and no model coupling (the dashed black lines).

Experiments 3 and 4 utilized a tighter coupling approach that enabled some feedback and non-linearity to take place (Table 1). Following the results of prior sensitivity experiments (Barton et al.,

2015), we repeated these two experiments 30 times. The result of tighter coupling and dynamic land-use produces a non-linear temporal trend in accumulated ED, with distinct inflections in the rate of ED change around years 70 and 90 of the simulation (Figure 2, b and c). When landscape changes cannot feedback to affect the social system (Figure 2-b), each set of parameters that initializes a human-land-use scenario generates a narrow range of variation in total accumulated ED simulated by LEM. With bi-

directional feedbacks between human and natural systems, however (Figure 2-c), both non-linearity of response and a wider range of variability among outcomes are produced.

*2.2 Effects of residential land management on carbon storage*

2.2.1 Definition and description

To quantify the effects of residential land management practices on ecosystem carbon, a framework was developed to couple the dynamic ecosystem model BIOME-BGC with a human decision-making model (Robinson et al. 2013). This coupling uses an agent-based design because it supports a conceptual model that includes different types of human agents acting in the system, their interactions, the scope of their

behaviours and influence, and their attributes (Rounsevell et al. 2012), and is therefore consistent with the conception of society as a complex system.

Our model of the human system, Dynamic Ecological Exurban Development (DEED) model, combines a suite of components developed to systematically incorporate additional data and complexity



in the residential development landscape (Brown et al. 2004, Brown and Robinson 2006, Brown et al. 2008, Robinson and Brown 2009, Robinson et al. 2013, and Sun et al. 2014). Farmer agents own land that is bid on by Residential Developer agents. The winning Residential Developer agent subdivides the landscape into one of three subdivision types, each with different residential lot density and land-cover

impacts (remove all vegetation, leave existing vegetation, grow new vegetation). Residential Household agents then choose a location for settlement and conduct land management activities.

The dynamic ecosystem model BIOME-BGC was used to represent deciduous broadleaf forest and turfgrass (i.e., maintained lawn) growth. It operates on a daily time step and reports outputs at daily and annual periods. Although the model was not developed to expose specific variables to land management,

it was selected because: 1) existing variables permit the representation of different types of vegetation growth found in exurban landscapes (Robinson 2012) like turfgrass (Milesi et al. 2005); 2) the parameters and inputs used by the model can be altered to represent the impacts of land management behaviours that affect vegetation growth; 3) the biogeochemical cycling in the model represents water, carbon, and nitrogen with extensive literature validating model outcomes, including parameterization for

different ecosystems and species (White et al. 2000); and 4) it has been applied both at high spatial resolutions (e.g., 30 m) and at local-to-global spatial extents (e.g., Coops and Waring 2001), which facilitates both the local site evaluation and the potential to scale out to regional or national levels.

### 2.2.2 Feedbacks

A loose coupling approach linking the ABM and BIOME-BGC was used to evaluate the range of impacts that land-management activities may have on carbon storage. Interaction between the two models was achieved through the exchange of information between output and input files. First, land exchange occurs and any land-use changes (e.g., residential development) are completed. Then,

Residential Household agents conduct land management activities. When all agents complete their activities, the ABM identifies those agents that input water or nutrients to the landscape. If irrigation is applied then the ABM modifies the climate input data used by BIOME-BGC for the year at that location by changing the amount of precipitation. If fertilizer is applied then the ABM alters the soil mineral

nitrogen in an initial conditions file (i.e., the restart file). Following these additions, the ABM modifies the schedule of BIOME-BGC and steps it forward by one year. The outputs from BIOME-BGC are then modified through removal activities. Removal of coarse woody debris and litter is achieved by reducing the respective variables in the initial conditions file for the subsequent year. After these management activities take place, the ABM framework summarizes a number of ecosystem variables for a given cell,

residential property, or landscape (e.g., total vegetation, litter, and soil carbon as well as net primary productivity) and with the implementation of carbon policies these ecosystem changes would affect subsequent agent land management activity choices.

Feedback from the ecological system on agent behaviour was explored through changes in policies that support offset payments for increased carbon storage. An alternative feedback could include effects

on social preferences and norms for landscape design elements (e.g., xeriscaping or adding tree cover) that may create social norms and drive changes in land management activities and subsequent ecological outcomes.

### 2.2.3 Case study

The model coupling was used to evaluate the effects of land-management strategies on carbon storage in exurban residential landscapes (Robinson et al. 2013). We constrained and initialized the ABM with a single farm parcel that is bid on and acquired from a Farm agent by a Residential Developer agent. The Residential Developer agent then subdivides the parcel into four identical low-





density (i.e., exurban) residential parcels (1.62 ha or ~4 acres) commonly found in southeastern Michigan, USA. Each parcel comprises a mixture of dense tree cover (i.e., broadleaf forest; 33%), turfgrass (56%), and impervious surface (11%). The site conditions are initialized, and BIOME-BGC is calibrated, based on ecological measurements from the region (Robinson et al. 2009).

Four Residential Household agents were created, and each locates in one of the four residential parcels. Each Residential Household agent has a unique set of land management activities. The four distinct management approaches are to do nothing (representative of no interaction or feedback between human and natural systems), remove course-woody debris from forest and litter from turfgrass, add fertilizer and irrigate turfgrass, or conduct both removals and additions. The coupled model was run for

48 time steps, which represents the years from 1958-2005, with land management activities carried out annually.

Results from the model experiments corroborate existing literature that forest cover stores significantly more carbon per unit area than many other land-cover types (Robinson et al. 2009, Robinson et al. 2013), in our case turfgrass, and therefore land management activities that alter land-

cover composition have the largest impact on carbon storage. Moreover, we found that land management that removes biomass from the landscape, in the form of litter or coarse-woody debris (CWD), had a larger effect on carbon storage than additions of fertilizer and irrigation. Furthermore, fertilizer and irrigation applied to turfgrass led to relatively little carbon gains over performing no management activities. Lastly, the variation in carbon storage resulting from different land management

activities on a relatively small amount of land (1.62 ha), was 42,104 kg C over 48 years, demonstrating that even the residential land management practices of individual property owners can have a substantial impact on carbon storage. While the magnitude of carbon storage and processing of other ecosystem functions will vary with climate and biophysical conditions, scaling these results to the extent of exurban

areas across the US, estimated to be 18% of the conterminous US (Theobald 2005), would further demonstrate the effects of land management on carbon storage.

Although the experiments above would have been possible without model coupling, the coupling permits further exploration of the effects of interventions on human behaviour (e.g., market incentives or

5     education) that can affect ecosystem function. Through the coupling process we identified which land management activities could be coupled to BIOME-BGC and different ways of representing land management activities (e.g., via alterations to climate data or initialization files).

### 2.3 Coupling vegetation, climate, and agricultural land use-trade models

2.3.1 Definition and description

To explore the interactions between land-use decisions, food consumption and trade, land-based emissions, and climate at a global scale, a dynamic global vegetation model (LPJ-GUESS, Smith et al., 2014), a land use and food system model (PLUMv2, Engström et al., 2016a), and a climate emulator

15     (IMOGEN, Huntingford et al., 2010) were coupled (3-i). Key objectives were: a) to represent the trade-offs and responses between agricultural intensification and expansion and the cross-scale spatial interactions driving system dynamics (Rounsevell et al., 2014), b) to explore whether climate and $CO_2$-related yield changes in a coupled system would affect projected land-cover change, and c) how these changes might feedback to the atmosphere and climate via the carbon cycle. A detailed representation of

20     yield responses to inputs (fertilisers and irrigation) was used and assumptions of market equilibrium were relaxed to allow exploration of the effects of shocks and short-term dynamics.

i)                                                   ii)

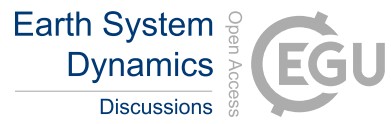

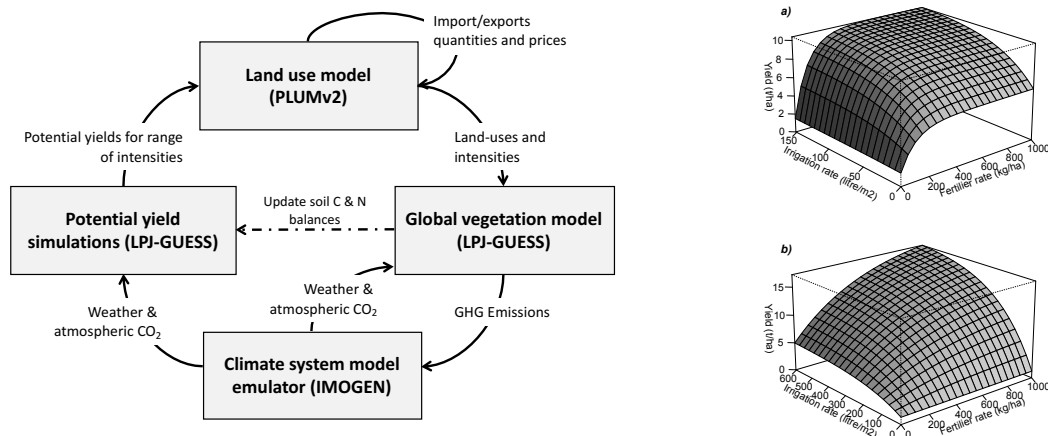

Figure 3. LPJ-GUESS, IMOGEN and PLUMv2 model coupling: i) diagram of main coupling interactions between these models; ii) example yield responses to fertiliser and irrigation inputs, used within PLUMv2 derived from LPJ-GUESS yield potentials at 2010, for wheat a) in England [51°N, 0°W], and b) in California [36°N, 120°W].

The carbon, nitrogen, and water cycles, as well as vegetation composition and ecosystem state, were simulated at 0.5-degree spatial resolution in LPJ-GUESS. Natural vegetation was represented using 11 Plant Functional Types (PFTs, e.g., Smith et al. 2014), whose growth was simulated by processes such as photosynthesis, respiration, and resource allocation. Competition between PFTs is based on gap dynamics (e.g. Smith et al. 2001), which captures natural succession within forests (Hickler et al., 2004;

Smith et al., 2014), for example following land-use change. Agricultural and pastoral systems are represented as a prescribed fractional cover of area under human land use per grid cell. Four crop functional types modelled on winter wheat, spring wheat, rice, and maize were used to simulate croplands (Olin et al., 2015), with representation of sowing, harvesting, irrigation, fertilisation, tillage, and the use of cover crops (Lindeskog et al., 2013; Olin et al., 2015; Pugh et al., 2015). Pastures are

represented by competing C3 and C4 grass, with 50% of the above-ground biomass removed annually to

represent the effects of grazing (Lindeskog et al., 2013). A general circulation model (GCM) emulator, IMOGEN, provides feedback between the carbon cycle and climate to be investigated without the computational demands of running a full Earth System Model (Huntingford et al., 2010).

Economic and behavioural aspects for country-level decisions within the food system were modelled in PLUMv2, extending Engström et al. (2016a). The PLUMv2 model projects demand for agricultural commodities based on socio-economic scenarios (e.g. SSPs, van Vuuren & Carter, 2014), and attempts to meet these demands through country level cost minimisation, including spatially specific land use selection, production intensity (fertiliser and irrigation rates), and the levels of imports and exports. Trade barriers, agricultural subsidies, and non-economic policy behaviours at a country level (such as

the introduction of export restrictions) are also represented. The approach uses the detailed yield data from LPJ-GUESS. Yield responses to changes in intensity of agricultural practices were represented on a 0.5-degree grid, across a range of irrigation rates and fertilisation rates (Figure 1). While local edge effects (e.g. Peters et al., 2007) are not explicitly simulated at this resolution, the computational demands become excessive at finer resolutions. Avoiding the use of a global equilibrium market assumption more

closely represents observed market behaviour and makes PLUMv2 suitable for modelling changes arising from shocks, e.g. extreme weather events. Shortfalls are met from stocks, with international market prices adjusted to bring the markets towards clearing.

### 2.3.2 Feedbacks

Yield potentials on a half-degree grid were produced by LPJ-GUESS for six combinations of fertiliser and irrigation rates based on climate in the previous five years. PLUMv2 fits exponential intensity response surfaces for each combination of grid cell and crop (e.g. Figure -ii) and uses these in the projection of land use and intensity, as outlined above. The resulting projected future land uses,

including fertiliser and irrigation rates for each crop, were then provided to LPJ-GUESS for the next time step, to allow soil carbon and nitrogen pools to be updated, and to calculate the net exchange of carbon between the global land surface and the atmosphere. This reflects land managers making decisions for the future based on climate and crop performance in the recent past. The land carbon

exchange is input to IMOGEN to simulate climate during the next time-step, and the coupling loop then repeats for the next time-period (e.g. Figure 3-i). The LPJ-GUESS-to-PLUMv2 coupling occurs on a five-yearly time-step, but both models internally use annual time steps.

### 2.3.3 Case study

The coupled LPJ-GUESS and PLUM models, without climate feedbacks from IMOGEN, were used to investigate land use change to the end of the 21$^{st}$ century using the SSP–Representative Concentration Pathways (RCP) framework and prescribed climate forcings. Calibration and validation against historical commodity demand and land use change data from 1970–2010 showed the model able to reproduce global demand, cropland and pasture area, and irrigation water and fertiliser use. In the future

land-use experiments, runs were conducted with climate forcings from RCP projections sampled based on conditional probabilities for each SSP scenario (Engström et al., 2016b). A Monte Carlo sampling approach was used to explore the effect of 18 parameters, including those related to technology change, trade barriers, and restrictions on land-use change. The results demonstrate that the multiple pressures on agricultural land within each scenario need not act in the same direction, but rather can act to partially

offset one another. Consequently, the area changes can converge as a result of different drivers. For example, the median change in cropland areas by 2100 was found to be an increase of 150–200 Mha in all the SSPs, except SSP5, in which a small reduction was shown (Rabin et al., 2017).

The LPJ-GUESS–PLUMv2 coupling allowed the geographically specific yield responses to intensity to be modelled, including how the trade-offs between inputs and yields vary between locations at the grid cell scale—with consequences for local strategies of adaptation to climate change. This small-scale variation can be missed in economic models where land use is determined at a regional or other

5   aggregate level, before disaggregation. The case study also demonstrates the ability to explore parameter uncertainty and interactions by running large number of simulations, given the ability to use high-performance computing, and with a relatively modest execution time.

### 2.4 The integrated Earth System Model (iESM)

2.4.1 Definition and description

The iESM (v1.0; Collins et al., 2015) directly couples the Global Change Assessment Model (GCAM, v3.0; Wise et al., 2014) with the Community Earth System Model (CESM, v1.1.2; Hurrel et al., 2013) to explore two-way, synchronous feedbacks between terrestrial ecosystems (including their interactions

15   with the climate system) and human land use and energy systems. This coupling follows the Coupled Model Intercomparison Project phase 5 (CMIP5; Taylor et al., 2012) Land Use Harmonization protocol (LUH; Hurtt et al., 2011), with some modifications and additions (Bond-Lamberty et al., 2014; Di Vittorio et al., 2014), and incorporates GCAM and the Global Land-use Model (GLM, v2; Hurtt et al., 2011) into the CESM framework via a newly developed integrated assessment coupler (Collins et al.,

20   2015). In the iESM, GCAM projects fossil fuel $CO_2$ emissions and land use every five years at the regional scale, which are passed through downscaling and land use translation algorithms for use by the atmosphere and land components of CESM. The non-$CO_2$ emissions are prescribed by CMIP5 data. CESM then runs for five years and the integrated assessment coupler uses the net primary productivity





and heterotrophic respiration outputs to calculate average crop productivity and ecosystem carbon density scalars, which are passed back to GCAM for use in projecting the next five years of land use and emissions (Bond-Lamberty et al., 2014).

GCAM is an integrated assessment model that represents both human and biogeophysical systems (Wise et al., 2014), although the climate and carbon components of GCAM are replaced by CESM in the iESM. The human components simulate global energy and agriculture markets to estimate anthropogenic emissions and land change. The solution for each time step is determined by balancing supply and demand for each commodity. The energy and land components are distinct, but connected via bioenergy, nitrogen fertilizer, and (where applicable) greenhouse gas emissions markets. Energy and agricultural commodity markets are modelled at the level of 14 geopolitical regions, and land change is modelled within 151 land use regions. There are 27 available land types, three of which remain constant (urban, tundra, and rock/ice/desert), and 24 that can be redistributed, including 12 food and feed crops, five bioenergy crops, and seven managed and unmanaged ecosystems. Land types are distributed based on profit shares from agricultural costs, prices, and yields in competition with the value of non-agricultural land, which can change if a carbon price is applied to ecosystem carbon.

CESM is a fully coupled Earth-system model with atmosphere, ocean, land, and sea ice components, including land and ocean biogeochemistry that exchanges carbon with the atmosphere (Hurrel et al., 2013). The standard resolution of all CESM components in fully coupled mode is nominally one degree, but the land cover is determined as fractions of half-degree grid cells and prescribed prior to a simulation (Lawrence et al., 2012). In the iESM, however, land cover is determined annually from GCAM, GLM, and the land use translator and passed to the Community Land Model v4.0 (CLM; (Di Vittorio et al., 2014; Lawrence et al., 2011). Biogeographical vegetation shifts are not included, although ecosystems do respond and contribute to changing environmental conditions. CLM includes detailed



hydrology and mechanistic vegetation growth for 16 Plant Functional Types (PFTs; eight forest, three

grass, three shrub, one bare soil, and one crop) in order to simulate water, carbon, and energy exchange

with the atmosphere.

2.4.2 Feedbacks

The two-way, synchronous coupling of GCAM's land change model and CESM's ecosystem model

implements feedbacks and generates outputs not available from the individual models or through one-

way coupling exercises. Within iESM, the atmosphere component and CLM receive $CO_2$ emissions and

translated land use change from GCAM every five years, and at the end of this interval CLM provides

terrestrial feedbacks to GCAM that incorporate changing environmental conditions (Collins et al.,

2015). These feedbacks are then used by GCAM to project the next five years of $CO_2$ emissions and

land use, thereby incorporating the effects of climate change, $CO_2$ fertilization, and nitrogen deposition

on terrestrial ecosystems into these projections.

These feedbacks provide new capabilities and information not available from the individual models.

The key new feature is the generation of vegetation and soil impact scalars from CLM data that are used

by GCAM to adjust crop productivity and carbon at each time step. This fundamentally alters the

scenario by making the land projection, and consequently the energy projection, more consistent with

the climate projection. The largest technical contribution, however, is the integrated assessment coupler

that enables feedbacks by running GCAM, GLM, and a new land use translator inline with CESM.

These capabilities enable new insights into research questions regarding climate mitigation and

adaptation strategies. For example, how may agricultural production shift due to climate change, how do

different policies influence this shift, and how may this shift affect other aspects of the human-Earth

system? Many recent impact studies (e.g., ISIMIP, BRACE, CIRA2.0) use climate model simulations



based on emissions and land-use scenarios (Representative Concentration Pathways, RCPs) that themselves do not account for the influence of climate change on future land use. This inconsistency could affect conclusions about impacts resulting from particular RCPs.

This approach paradoxically has several strengths that are also weaknesses. The main strength of this approach is that it couples two state-of-the-art global models to implement primary feedbacks between human and environmental systems under global change. Unfortunately, this configuration is not amenable to the uncertainty and policy analyses or the climate target experiments usually employed by GCAM because it takes too long to run a simulation. As a global model it provides self-consistent representation of interconnected regional and global processes, both human and environmental, but is unable to capture a fair amount of regional and local detail that influences planning and implementation of adaptation and mitigation strategies. Furthermore, it was based on the CMIP5 protocols to facilitate its development and comparison with existing coupling approaches, but these protocols do not require consistency between models, which is critical when models are coupled directly.

### 2.4.3 Case study

The first iESM experiment using RCP4.5 demonstrated that the inclusion of terrestrial feedbacks in GCAM dramatically altered land use and fossil fuel forcings to the climate system, with subsequent effects on the global carbon cycle (Thornton et al., 2017). In this experiment, the $CO_2$ emissions were prescribed by CMIP5 data in order to examine the first order feedback without additional effects on ecosystems due to a different emissions profile. Two historical, one-degree simulations starting from two different, pre-industrial equilibrium conditions in 1850 provided two sets of initial conditions for three RCP4.5 simulations. Only CESM was active for the historical simulations, which used the CMIP5 prescribed land use and emissions. One control simulation (no terrestrial feedback) was compared with

two feedback simulations from 2005 to 2094 to quantify the effects of introducing feedbacks into the human-Earth system.

Including terrestrial feedbacks enabled adaptation to environmental change, thus altering the RCP4.5 scenario with respect to land use, bioenergy production, and fossil fuel emissions. The land use

5    differences increased terrestrial carbon and decreased atmospheric carbon, with significant changes in regional distribution of terrestrial carbon. Forest area and bioenergy production increased, and fossil fuel $CO_2$ emissions decreased by 17% in 2100. Understanding this potential adaptation and how it may affect human-Earth system projections would not be possible without implementing feedbacks through the coupling of human and Earth-system models.

## 3. DISCUSSION

The examples presented here demonstrate how specific models of human and natural processes,

15    developed by specialists, have been connected through different coupling approaches to address research questions related to the impacts of one system on another and the effects of feedbacks between human and natural systems for a variety of variables of interest (e.g., erosion, carbon storage and emissions). These examples provide a level of transparency and detail in the represented processes that is not typically found in larger Integrated Assessment Models (IAMs). This not only facilitates our capacity

20    and ability to answer new types of questions about coupled human-natural systems, but also fosters the identification of new types of data about the interactions and feedback pathways between the two systems.

The example models also illustrate diversity in the spatial and thematic resolutions of human and natural-system model coupling. The first two examples (MML and DEED) use agent-based approaches that represent land use and land management in the human system at the household level. While the ecological impact of land management activities in DEED does not have a direct feedback on residential

household decision-making, those represented in MML do. Agricultural systems, carbon markets and policies provide mechanisms to establish this feedback and endogenize the impact-response cycle between the residential land management practices of the human system and the natural system (Sun et al. 2014).

The second two examples are global models with different levels of coupling and complexity that

represent human-natural system feedbacks at regional and gridded levels. In both systems, the human system has a direct effect on modelled Earth-system processes (i.e., vegetation, carbon, climate), and the feedback of environmental changes on human systems is mediated by vegetation responses to changing conditions.

The range of applications demonstrates a move toward coupled representation of human and natural

processes that are designed to extend the questions that models can be used to answer. The types of research questions to which coupled human-natural system models are able to contribute include quantifying the impacts of one system on the other with and without feedbacks present (e.g., What is the effect of the natural system on human system?), quantifying the relative impact of a perturbation or scenario on each system (e.g., do feedbacks dampen or amplify the outcome of the perturbation?),

identifying thresholds (e.g., what is the critical value of a variable in one system that when crossed instantiates change in the other system?), and resilience, robustness, or sensitivity related questions (e.g., what is the capacity of one system to absorb changes in the other and over what time period?), among many others.

### 3.1 Lessons Learned

To make methodological and scientific progress on understanding coupled human-natural systems, the variable or process of interest needs to be characterized in both systems. For example, precipitation has a known and direct effect on plant growth (e.g., forest or crop) and erosion (e.g., overland flow). The outcome of these processes (e.g., yield and soil loss) have a direct or indirect effect on land management choices by farmers, which is known at least through qualitative understanding if not quantitatively measured. However, the direct impacts of other perturbations, such as the introduction of new technology, on either of the human or natural-system processes are more difficult to identify. For this reason, we see in the presented case studies research questions that accompany perturbations or scenarios that are grounded in known and direct causal factors. The factors of focus are more likely to be found in the natural system than the human system due to the multitude of drivers affecting—and consequent difficulty in predicting—human-decision making. Nonetheless, there are substantial knowledge gaps across both systems associated with human-environment interactions.

One advantage of coupling human and natural system models is that it often helps with the identification of key variables, data, or mechanisms that need further investigation. For example, through the construction of the DEED model (Section 2.2), vegetation and soil carbon data in residential land uses were identified as needing improvement, which fostered new data collection and findings about the distribution of carbon stored in different residential land uses (Currie et al. 2016). New forms of measurement and evaluation may be needed to quantify feedbacks between and relative effects of perturbations in human versus natural systems



Another advantage of model coupling is that gaps in process representation in models can be filled through judicious choice of parameters, use of simulated data, or expert- or theory-informed methods that enable the model to be leveraged to evaluate the relative contribution of missing data or methods to model outcomes. For example, in the absence of social network data, Agrawal et al. (2013) evaluated

changes in fuelwood extraction and forest growth between different household networks ranging from adjacent neighbours to distant ties. The outcome of such efforts can be used to determine if real-world effort and resources should be deployed to capture the process empirically and provide scope of sample design.

On the other hand, coupling models increases computational overhead and thus requires increases in

computational efficiency, which always comes with trade-offs. One approach to improve efficiency is to classify and generalize components of the model such as agent types in the human system (e.g., Brown and Robinson 2006), types of vegetation (e.g., plant functional types, Díaz and Cabido 1997, Smith et al. 1993, Smith et al. 1997), or landscape units. The landscape unit is not typically used to structure spatial variability in land use science, but is used regularly in hydrological modelling (e.g., SWAT,

Neitsch et al. 2011) in the form of hydrological response units (HRUs) that have a soil profile, bedrock, and topographic characteristics that are assumed homogeneous for the entire spatial extent of the unit. Similar concepts have been used to identify management zones or units, and two of our examples employ this approach (Robinson et al. 2013, Collins et al., 2015). However, the variability between management activities and land cover types can lead to a large combination of outcomes, and the

delineation of these units directly contributes to projection uncertainty (Di Vittorio et al., 2016). Such compromises are expected to have a minimal effect at the global scale, but locally the effect may prove to be substantial, and is challenging to evaluate.



### *3.2 Consistency*

Because natural-systems models and human-systems models sometimes attempt to simulate similar phenomena, like land cover, coupling existing models can encounter significant challenges in order to

maintain ontological and process consistency. Models with different initial assumptions and different processes can generate different values for the same phenomenon. While model coupling ultimately can help to harmonize and resolve such consistency issues, it requires decisions about which processes to represent and which to leave out in the coupling procedure to avoid duplication.

The iESM (Section 2.4) well illustrates issues of consistency in assumptions, definitions, and

processes. First, ecosystem properties from CLM were translated to impacts that could be applied to GCAM "equilibrium" yields and carbon densities (Bond-Lamberty et al., 2014). Second, a major finding that is especially relevant to all land change and ecosystem models is that the inconsistencies between a land use approach and a land cover approach caused CESM to include only 22% of the prescribed RCP4.5 afforestation in CMIP5 (Di Vittorio et al., 2014). Additionally, it was discovered that wood

harvest was conceptually different across three of the models comprising iESM (GCAM, GLM, and CLM), with each model having a unique carbon cycle determining how harvest is spatially distributed. Wood harvest is a good example of where the different modeling groups thought that they were all describing the same thing, partly because the global harvest amounts were similar across models, but actually very different things were happening in each model, with unintended consequences for CESM's

terrestrial carbon cycle.

The DEED example (Section 2.2) demonstrates issues of temporal consistency. Reconciling the temporal mismatch involved balancing detail and computational tractability with existing model structures. The DEED ABM used an annual time-step to reflect the pace of land change, whereas the



ecosystem model BIOME-BGC used daily weather inputs. Irrigation decisions were made annually, but implemented one day a week during the growing season by modifying the daily precipitation file. In contrast, other management activities were implemented once annually, before (for fertilization) and after (for removals) the growing season. These limitations could have a significant effect on estimated

carbon storage and have fostered additional fieldwork for further validation (e.g., Currie et al. 2016) and additional efforts to tightly couple the two models.

A third consistency issue involves the balancing of equations and acceptability of variable and parameter values. Many ecosystem and Earth-system models have mass, energy, or other balance equations that establish the system as a closed system. These equations ensure models abide by the first

law of thermodynamics, whereby the total energy in a closed system is constant and therefore cannot be created or destroyed. For example, the ecosystem model BIOME-BGC has a routine to ensure balance between influx, efflux, and storage of carbon, water, and nitrogen. Similar checks and balances are used in some human system models with respect to population change (e.g., births, deaths, immigration, and emigration) or economic trade (e.g., production, consumption, imports, and exports) at macro levels and

budget or labour constraints at household or individual levels. However, in many natural-system models these balance equations are not accessible for coupling and the representation of human perturbations (e.g., carbon additions and removals through land management in the DEED example above) and modifications to the factors in balance equations are either not included or done so indirectly.

Another complicating factor is that some models are strongly deterministic, so that the results are

essentially the same for any run with the same initial parameters, while others have strong stochastic components. The former is common for many natural-systems models and some human-systems models (especially econometric style models). Other models have algorithms that generate stochasticity to represent uncertainty in processes. Many agent-based/individual-based models and some cellular

automata fall into this category. For models with included stochasticity, using the same random number

generator seed can be used, however, to capture the variance and distribution of model outcomes

repeated runs with different seeds are required, which can be conceptually challenging or operationally

complicated when coupled with deterministic models.

### 3.3 Feedback Effects

Introducing feedbacks often changes model outcomes, but it is difficult to determine if these changes are

significant relative to the uncertainty associated with inconsistency between the original models. The

only way to test if the representation of a mechanism concurs with observed behaviour in the real-world,

is to remove the inconsistencies so that a comparison can be made. In this case, the actual feedback

effects would be quantified. Barring this level of model development, one must hypothesize that if

feedbacks change model outcomes, then potentially significant feedback dynamics exist regardless of

model consistency. Then useful feedback experiments can be carried out and evaluated, as long as the

coupling and associated dependencies are clearly defined.

Such model experiments can facilitate quantification of feedback effects, which provides

guidance and insight into processes that are typically not observed and measured or may be impossible

to measure (in some cases) in the real world. Given the difficulty in observing these effects and potential

inconsistencies, efforts in coupling human and natural-system models have focused on exploring the

range of outcomes achieved with and without coupling (i.e., the addition and removal of feedbacks),

which offers a form of sensitivity analysis (e.g. Harrison et al., 2017). This gradient of coupled-model

instantiations offers an ensemble approach to understand the boundaries of individual human and natural-system responses to tested perturbations or scenarios.

For the local model examples (MML and DEED), the changes in the natural system were relatively linear when human perturbations were made to the natural system and when feedbacks were not

included between the systems. However, when feedbacks between the systems were incorporated in a loosely coupled framework then non-linear outcomes were observed. The loose coupling offered a general idea of the boundary extents of model results, which provides guidance for tighter coupling and further experimentation.

The overall effects of feedbacks on model outcomes can come from two distinct sources: 1) model

implementation (both technical and conceptual), and 2) the actual feedback signal. Technical implementation often depends on conceptual implementation, which is related to the level of consistency between the original models. For example, the need to translate changes in gridded patterns of ecosystem productivity to changes in regional average carbon density can lead to varying sensitivities across magnitudes of productivity and areal change, requiring the unrealistic sensitivities to be filtered

out (Bond-Lamberty et al., 2014). The actual feedback signal can also be an aggregate of expected, direct effects and additional, indirect effects. For example, average ecosystem productivity could change due to atmospheric influences (e.g., climate change, $CO_2$ fertilization) or to terrestrial influences associated with changing ecosystem area (e.g., spatially heterogeneous soil properties, changes in the proportion of different forest types).

The iESM example demonstrates how model experiments are useful for characterizing both of these sources of feedback effects. For the iESM, Bond-Lamberty et al. (2014) used a series of CLM-only simulations to (a) quantify the relationship between ecosystem productivity and carbon density, (b) implement a statistical method to remove outliers that introduce error due to extreme combinations of



land cover and productivity change, and (c) develop the appropriate proxy variables for use by GCAM. In order to verify that this process effectively removed the majority of feedback effects associated with model implementation, another series of CLM-only simulations was conducted using the iESM with and without terrestrial feedbacks, and with constant atmospheric conditions (i.e., year 1850 aerosols and

nitrogen deposition, and repeating 15 years of climate forcing). This experiment showed that the coupling itself, without an atmospherically driven feedback signal, did not generate significant changes in GCAM outputs (unpublished). The feedback signal was not zero, however, indicating that a combination of implementation effects and land heterogeneity effects was present. These combined effects were not separable due to lack of the required outputs, and in general they were opposite to the

total feedback effects in the fully coupled iESM experiment, suggesting that the atmospherically driven feedbacks may be larger than the net feedbacks.

While feedback mechanisms may have large consequences on the behaviour of the coupled Earth system, they may have similar impacts on coupled model behaviour. Depending on the mechanisms involved they may create strong feed forward effects that lead to fast evolution of the dynamics of the

system. On the contrary, responses may also lead to an attenuation of the dynamics and strong stability of the dynamics. Where the exact mechanisms involved and the strength of the feedbacks are unknown, model dynamics may start deviating strongly upon small changes in variable settings, in other words, leading to strong model sensitivity to highly uncertain model parameters.

The experiments including feedbacks in coupled model systems illustrate a number of challenges in

designing and implementing feedback mechanisms:

1)  Spatial and temporal consistencies: both the processes addressed and their respective model representations may operate at different spatial and temporal scales. In coupling of models this may lead to mismatches such as illustrated by the ABM-BIOME-BGC coupling. Often, these



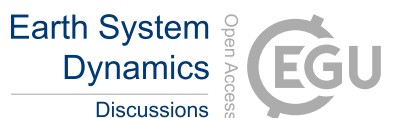

cannot simply be dealt with by aggregation of the variables as the mismatch in temporal and spatial dynamics may also happen in reality. Human responses to environmental change may show significant time-lags or may be related to cycles of management (e.g. cropping cycles) rather than showing an immediate response. Similarly, while the ecological models are strongly

place-based, coupling human and natural systems at the pixel level may not always be appropriate due to complex spatial relations in the human dimensions (e.g. distant land owners) or responses across different levels of decision making (e.g. policy responses) that are not linked to the exact place of the ecological impact.

2)   The representation of human responses: the examples in the four cases above mostly relate to a

coupling based on exchanging land cover and ecological-process-impact information. The human decision models translate the ecological impacts to changed decision making. For example, in the PLUM-LPJGUESS model, land-use decisions respond to changes in potential yields and in the MEDLAND application erosion processes render land less suitable for use. In reality the responses of human decision-making are more complex. The relevance of the

ecological indicator exchanged may be context dependent, i.e., potential yields may determine farming decisions in capital intensive farming that is near to the production frontier, but be much less important in low-input subsistence farming that is far from potential productivity. Furthermore, decision-making may not be based on the represented process or by the indicator exchanged, but rather on the human perception of the environmental change, which may be

irrational and biased by other interests, such as in case of the climate debate. While the concept of environmental cognitions is well known, relatively little is known in relation to land-use change decision-making (Meyfroidt 2013). Human responses to environmental change are, therefore, a critical knowledge gap for implementing coupling mechanisms (Meyfroidt 2013b).

3) Structural differences in model concepts: The iESM example nicely illustrates how structural
differences in model concept can hamper the coupling of models, where careful consideration is
needed on the feedback mechanism and its consequences in relation to the overall model
assumptions. This is especially relevant for the coupling between models that assume
5    equilibrium and those that depict instantaneous impacts or a transient situation. Both global
economic models using equilibrium assumptions, which are frequently coupled to land use and
ecosystem models, and specialized land-change models (or land-change modules in IAM
models) address land use, but often from different perspectives leading to potential differences in
the meaning and interpretation of exchanged variables.

## 4. CONCLUSIONS

Coupling human-system and natural-system models requires connecting two distinct research fields,
15    each with unique knowledge, methods, assumptions, definitions, and language. Success depends on the
research team members learning enough about the other field and model to unambiguously
communicate with each other, recognize strengths and weaknesses of other methods, translate
assumptions and definitions, and critically evaluate other model processes and outputs. Additionally,
some team members need to develop working expertise of both models and fields to facilitate
20    implementation of an internally-consistent coupled model. Furthermore, a team member with training in
computer science is often needed to address the technical challenges of coupling complex models. All
authors noted that the greatest benefit of the coupling process was the collaborative learning process that
created a group of people with working knowledge of both human- and natural-system research and

expertise in how to integrate the two. Ultimately, the social and conceptual challenges combined to require much more time and effort for project completion than originally allocated.

While successful coupling of human and natural-systems models requires truly interdisciplinary collaboration, we note that the playing field is not level with respect to disciplines. There are more

resources and, hence, active modeling efforts in the natural sciences than in human systems science. This is unfortunate since Earth scientists need to closely work with human systems scientists to understand the kinds of information needed and the kinds of information that can be provided by models of human systems. Moreover, the most scientifically and socially valuable results of model coupling require that both Earth-systems models and human systems models be modified and enhanced to work

together. The collaborative model development that this entails involves social interactions, two-way communication, and mutual respect for needed domain knowledge as well as technical solutions. In this regard, there needs to be scientific, professional, and policy incentives for all members of the interdisciplinary teams needed to develop successful integrated modeling.

These efforts highlight the difficulty and challenges associated with the process of human-

environment model coupling as well as the opportunities that coupling presents for making substantive and methodological advances in science associated with human systems, natural systems, and their feedbacks with each other.

**5. ACKNOWLEDGEMENTS**

This research has been made possible for the authors from a variety of supporting institutions, which we thank and acknowledge in what follows. DR was supported by the Natural Sciences and Engineering



Council (NSERC) of Canada as part of their Discovery Grant program. AD was supported by the U.S. Department of Energy, Office of Science, Office of Biological and Environmental Research, Climate and Environmental Sciences Division, Integrated Assessment Program, under Award Number DE-AC02-05CH11231. PA, TP, and MR were supported by the Future Earth AIMES project, CSDMS and

the European Commission LUC4C project. AA, TP, SR, and MR also acknowledge LUC4C (grant no. 603542) and the Helmholtz association through its ATMO programme and its Integration and Networking fund. CMB and IU were supported by the US National Science Foundation (grants BCS-410269 and DEB-1313727), and support from Arizona State University and the Universitat de València, Spain. Many other people in Jordan, Spain, and the US contributed in various ways to the MedLanD

project and we want to extend our thanks to them also. AK and JS were supported by the CSDMS project, funded by The US National Science Foundation (grant 0621695). CL was supported by the MOSSCO project funded by the German Ministry of Education and Science (BMBF) under grant agreements 03F0667A. BO contribution was based upon work supported by the National Science Foundation under Grant Number AGS-1243095. PV was supported by European Union's Seventh

Framework Programme ERC Grant Agreement nr. 311819 – GLOLAND.

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
