# Peer review of "Modelling feedbacks between human and natural processes in the land system"

_Earth System Dynamics, 2017_

## Referee Comment (RC1) · Anonymous Referee #1 · 14 Aug 2017

This article analyses feedbacks between human and natural processes in land system models. The author describe in detail four recent models of this type and then attempt to draw general conclusions.

I do not have the expertise to comment on the technical aspects of the manuscript. At a more conceptual level, the paper does not seem particularly ambitious. It is mostly descriptive in nature and not particularly analytical. As judged by the abstract, for example: the abstract is mostly a general introduction and then a vague summary of the article structure. I would like to see emphasised more clearly WHAT sorts of "lessons" are learned or "challenges" discussed rather than just that the statement THAT these things are done in the manuscript.

The bulk of the manuscript is a detailed description of the models and some illustrative

results. I'm not quite sure what role the results are meant to serve. There is no space to expand on them in detail, and they are not picked up upon in the discussion.

In section 3.1, "Lessons learned" the discussion is actually rather general. Only one model (DEED) is actually mentioned. I would like to see more discussion of what was learnt from the models that were discussed in so much detail earlier in the manuscript.

I'd also like a broader discussion of research gaps. Although combining human and natural processes is an important subject, the models discussed here still only address a subset of human processes. In section 2.3, for example, societal and cultural level processes of norm formation regarding food consumption for example are discussed. How else to describe changes in food consumption patterns? Such models probably don't exist in any form, and so can't be expected to be included in this paper, but the authors should indicate that there are still significant research gaps out there.

Minor comment: page 20 line 5: which land management activities were identified?

---

## Referee Comment (RC2) · Anonymous Referee #2 · 4 Oct 2017

The authors present a review of 4 approaches to representing interactions between humans and the environment in land systems using coupled models. Based on the review of these approaches, the authors provide discussion and recommendations for representing human-environment interactions in land systems using coupled models.

The subject of the article is interesting but work is required to restructure the manuscript, improve writing style and provide novel insights based on the review provided.

Comments

1. The 4 approaches are presented in an inconsistent way in Section 2.

a. My recommendation is that in section 2, in each case, introduce the type of model

coupling you are referring to, the motivation of the original researchers for coupling the models and how they are coupled. Following this, as you have done, outline the inter-actions/feedbacks each model captures between human and natural systems. A case study is useful, as you have done. I recommend following the case study with a sub-section explicitly outlining the strengths and weaknesses for each of the 4 approaches. These should be summarised and combined in a table in the discussion section of the document so the reader has a clear impression of how each of the 4 approaches com-pare. A diagram illustrating the differences in coupling or how feedbacks are captured may be instructive.

b. Section 2.1 and 2.2 headings sound like case studies and therefore this should be the sub-section title related to case studies. Since you are comparing 4 approaches, the main title of section 2.1, 2.2, 2.3 and 2.4 should illustrate the approach you are addressing in each section as you have done in section 2.3 and 2.4. c. The cases provided in section 2.1 and 2.2 are new models or approaches. As a review paper, I find this somewhat problematic as the idea, I would think, is to appraise pre-existing ap-proaches rather than adding two new approaches. If there are pre-existing approaches that are consistent with those presented in section 2.1 and 2.2, then these should be reviewed. If the authors feel that the approaches they are presenting here are novel and address some of the shortcomings of the approaches in section 2.3 and 2.4, then by all means present your approaches as novel frameworks. However, if they are not, then this article does not strike me as the appropriate platform to present new work (as it is now structured).

2. The article states that there is a lack of suitable frameworks to guide the build-ing of coupled models of land systems. Given the wide-ranging expertise of the co-authors here, I would expect the article to present a new framework to tackle these issues based an assessment of the strengths and weaknesses of existing approaches. I understand that you are addressing systems that operate across multiple scales. Nonetheless I would like to see the authors make an attempt to outline a framework

that can be applied across scales or at the minimum a list of best practices and core knowledge gaps that are required to be filled. A call to action, if you will. These should be explicitly outlined as they will form the main novel findings of the manuscript.

3. The writing style is very loose with statements made that are imprecise, insufficiently explained or qualified. For example, the motivation for coupling on page 5 line 16 – page 6 line 7. Page 6 line 16-17. The coupling approach enables a greater degree of transparency and accuracy in coupled models. There are examples like this throughout the manuscript that need to be made more precise, properly qualified and backed up by citations where appropriate. In addition, many of the sentences are too long and contain multiple arguments. Better to split these into shorter sentences for clarity.

4. The motivation of the study (stated page 6 line 17 – page 7 line 5) is unclear to me. These are the aims are written as:

Aim 1: We present multiple approaches to coupling land-change and natural-system models to evaluate how alternative approaches to representing feedbacks add value to scientific inquiry into global change and thus generate new insights into the sustainable management of human-environment interactions.

Aim 2: Based on the current state of the science, we categorize conceptual approaches to coupling land-change models with natural-system models that differ across a range of spatial extents and coupling methods.

Aim 3: Using four case studies, we critically assess the influence of land-change processes on natural-system processes and vice versa, focusing on the implications of these feedbacks for system dynamics, the research questions that model coupling enables, and the strengths and weaknesses of the coupling approach.

Aim 4: we describe the lessons learned from the various approaches, the different types of consistency that should be maintained between coupled models as well as the feedbacks represented between the human and natural systems.

Aim 1: I thought the focus was on the land system and land system models, not global change, which is a much broader topic.

Aim 2: Okay, a useful aim but I don't think the paper achieves this. You should make a clearer distinction of the differences between the 4 approaches presented. See comment 1 and 2.

Aim 3: I thought the specific aim of the paper was to investigate how human-environment interactions in the land system are captured using coupled modelling approaches. As it is written, it comes across as quite a verbose research aim. I recommend focusing solely on how human-environment interactions in the land system are captured using coupled modelling approaches.

Aim 4: See comment 1 and 2.

Specific comments

Page 4: Avoid the use of footnotes

Page 6 Line 20: State of the science is incorrect English

Page 28 Line 18-19. Please qualify this statement.

Page 38 and 39. The conclusion section on interdisciplinary collaboration is valuable and insightful.

---

## Author Comment (AC1) · 2 Nov 2017

Earth System Dynamics Discussion Response to Reviewers

Responses from Robinson et al. incorporated.

We would like to first thank the reviewers and the editor for their insightful comments that have pushed us to produce a higher quality paper. Thank you.

Reviewer 1

This article analyses feedbacks between human and natural processes in land system models. The author describe in detail four recent models of this type and then attempt to draw general conclusions.

[Figure]

I do not have the expertise to comment on the technical aspects of the manuscript. At a more conceptual level, the paper does not seem particularly ambitious. It is mostly descriptive in nature and not particularly analytical.

RESPONSE

There is an important lack of integration of social science in studies of earth system sciences. Living in the Anthropocene, and trying to better understand processes of this epoch and its interactions, we should even more so define ways to connect modeling activities of social science and natural science models. This by itself is ambitious, as there are major challenges that require different disciplines collaborating with each other, scientists need to adjust their numerical models so they can communicate with each other, and we need to learn a sufficient amount of domain knowledge and perspective from each other's models to perform an integrated analysis. In the current academic incentive system this kind of collaboration is rarely rewarded.

In response the reviewer comment above, a majority of the manuscript authors responded with statements that to the best of their knowledge this is the first paper to directly address the value of modelling feedbacks between human and natural systems for representing the dynamics of the socioecological systems that now dominate our world. In most current modelling efforts these feedbacks are largely ignored.

Collating the presented four cases into a single narrative offers a unique opportunity to observe the challenges and approaches taken to couple models developed by specialists in disparate academic fields. These case studies illustrate successful research outcomes that can provide building blocks and guidance to model coupling in the future.

We agree with the reviewer that the manuscript illustrates the current capability of representing feedbacks between specialized human and natural system processes, but this in itself demonstrates significant scientific advances and progress towards community building across disciplines. However, given the challenge set forth by the reviewer to be more ambitious we

1) Added a figure that provides a conceptual outline for the structure of coupling (Figure 1) and added a figure that illustrates different levels of coupling along a gradient of communication frequency and coordination (Figure 2). While these two figures do not cover all configurations of model structures or levels of communication frequency and types of coordination, they provide a conceptual outline that we use to describe the four examples in the paper and offer them as opportunities for other to do the same and contribute their work as comments to this manuscript or build upon and extend in subsequent manuscripts.

Figure 1: Approaches to model coupling. a) loose model integration via file / data exchange between model 1 (M1) and model 2 (M2); b) models share inputs and outputs but interact with independent data; c) models interact with the same data and share inputs and outputs directly with each other; d) a coupler coordinates run time and scheduling and may pass some information between models, models may also interact through manipulating data (files); e) a coupler coordinates the run time and scheduling of the individual models and passes information between models primarily use their own data, f) the coupler coordinates all interactions between models and data.

Figure 2. Conceptual outline of the frequency of model communication and coordination of interaction between models from no coupling to one-way and two-way feedback. Examples are not exhaustive but illustrate common approaches used. M1 = model 1, M2 = model 2, T1 = time step 1, Tn = time step n.

2) We also provided an additional subsection to the Discussion section that outlines 8 explicit lessons learned and a framework for coupling as guidance for a way forward.

REVIEWER COMMENT

As judged by the abstract, for example: the abstract is mostly a general introduction and then a vague summary of the article structure. I would like to see emphasised more clearly WHAT sorts of "lessons" are learned or "challenges" discussed rather than just that the statement THAT these things are done in the manuscript.

RESPONSE

Agreed. The abstract could better emphasize the insights and findings of our manuscript rather than providing a summary of what resides within. We will list and provide context for our 8 lessons learned

Lesson 1. Leverage the Power of Sensitivity Analysis with Models Lesson 2. Modelling is an Iterative Process. Lesson 3. Ensure Consistency Lesson 4. Reconcile Spatio-temporal Mismatch Lesson 5. Create a Common Language Lesson 6. Construct Homogeneous Units Lesson 7. Make Code Open-Access Lesson 8. Incorporating Feedback Increases Non-Linearity and Variability

We will also include text about the challenges faced in coupling and a short outline of how we suggest a way forward. Thank you.

REVIEWER COMMENT

The bulk of the manuscript is a detailed description of the models and some illustrative results. I'm not quite sure what role the results are meant to serve. There is no space to expand on them in detail, and they are not picked up upon in the discussion.

RESPONSE

Our intention was to draw upon the results as illustrative of achievements that could not be made in the absence of coupling models together or our approaches to model coupling. These results provide tangible outcomes for reference in the lessons learned section. While we had mixed response among our author listing about inclusion of the results in the manuscript versus relegating them to the supplementary material, we have decided to align with the reviewer and have created a piece of supplementary material that contains the results for each of the model coupling examples presented.

REVIEWER COMMENT

In section 3.1, "Lessons learned" the discussion is actually rather general. Only one

model (DEED) is actually mentioned. I would like to see more discussion of what was learnt from the models that were discussed in so much detail earlier in the manuscript.

RESPONSE

Earlier drafts of the paper had the lessons learned after each example and we attempted this again; however, in both attempts we collectively agreed that the lessons were more useful and effective as a group in the discussion section. Therefore we left the lessons in the discussion but removed the consistency section and refocused the lessons learned to make them explicit and numbered. Each lesson also now refers back to and offers an example from one of the four models presented. Our draft of this section is as follows: 
[revised manuscript text omitted]
. Results from the four examples provided span the supplementary material and a number of publications. Among these coupling efforts, it has been found that the incorporation of two-way feedbacks (Figure 2) between models of the human and natural system typically produces non-linear results and a greater range in model outcomes than are observed when the models are isolated or one-way prescriptions are used. For both the MML and DEED models, changes in the natural system were relatively linear when one-way human perturbations were prescribed. However, when feedbacks between the systems were incorporated then non-linear outcomes were observed and frequently a greater variation in model outcomes (e.g., Supplementary Material 1.2).

REVIEWER COMMENT

I'd also like a broader discussion of research gaps. Although combining human and natural processes is an important subject, the models discussed here still only address a subset of human processes. In section 2.3, for example, societal and cultural level processes of norm formation regarding food consumption for example are discussed. How else to describe changes in food consumption patterns? Such models probably don't exist in any form, and so can't be expected to be included in this paper, but the authors should indicate that there are still significant research gaps out there.

RESPONSE

The reviewer addresses an important point, what are the research gaps, when integrating human and natural processes. However, coupling existing models does not necessarily mean that all processes are included for each domain. Indeed there are major elements of social systems or natural processes that are not well represented in available and existing models.

Here we focus on land use because it is a widely studied human-environment relationship that offers some examples of human-environment coupled models, which is the emphasis of this manuscript. Thus, we demonstrate how the authors have layed

the groundwork for increasing the complexity in subsystem models. The case studies demonstrate how the inter-system coupling is extremely important in addition to how the inter-system coupling affects estimates and our understanding of the coupled system. Once we know the implications of coupling models, then we can start coupling different types of models, and social scientists can create models of dynamics that have not been represented before, and connect it with natural system models (e.g., models of energy or transportation systems with climate models) and visa versa.

Through the manuscript we try to emphasize that there is considerable value in coupling models of other dimensions of human systems, created by social scientists, with relevant biophysical models. We do not seek to discuss different kinds of models of human decisions and actions. Many such models already exist (e.g., see model library at www.comses.net). However, most of these models have not been coupled with models of natural systems such that that they impact and are impacted by the natural system. The main focus of this paper, which we intend to clarify, is to show what new insights can be gained when we create modeling environments that can dynamically simulate interactions and feedbacks between different components of human and natural systems.

REVIEWER COMMENT

Minor comment: page 20 line 5: which land management activities were identified?

RESPONSE

Thank you for drawing attention to the mentioned text. We have revised revise the sentence to include the following land management activities: irrigation, fertilization, biomass removal.

Again. Thank you for your time and effort in providing a very helpful review.
* * *
[Figure]

**Fig. 1.** Figure 1: Approaches to model coupling.

[Figure]

**Fig. 2.** Figure 2: Conceptual outline of the frequency of model communication and coordination of interaction between models from no coupling to one-way and two-way feedback.

---

## Author Comment (AC2) · 2 Nov 2017

Earth System Dynamics Discussion Response to Reviewers

Responses from Robinson et al. incorporated.

We would like to first thank the reviewers and the editor for their insightful comments that have pushed us to produce a higher quality paper. Thank you.

Referee #2

The authors present a review of 4 approaches to representing interactions between humans and the environment in land systems using coupled models. Based on the review of these approaches, the authors provide discussion and recommendations for representing human-environment interactions in land systems using coupled models. The subject of the article is interesting but work is required to restructure the manuscript, improve writing style and provide novel insights based on the review provided.

REVIEWER COMMENT

1. The 4 approaches are presented in an inconsistent way in Section 2.

a. My recommendation is that in section 2, in each case, introduce the type of model coupling you are referring to, the motivation of the original researchers for coupling the models and how they are coupled. Following this, as you have done, outline the interactions/feedbacks each model captures between human and natural systems.

A case study is useful, as you have done. I recommend following the case study with a subsection explicitly outlining the strengths and weaknesses for each of the 4 approaches. These should be summarised and combined in a table in the discussion section of the document so the reader has a clear impression of how each of the 4 approaches compare. A diagram illustrating the differences in coupling or how feedbacks are captured may be instructive.

RESPONSE

We appreciate the pressure applied on us by the reviewer to create additional diagrams illustrating the differences in coupling or how feedbacks are captured. As mentioned in our response to Reviewer 1, we have created two conceptual figures (Figures 1 and 2) and included them in our introduction to coupling at the front of manuscript. In addition to these diagrams/figures, each of the four examples now relates back to these conceptual figures and identifies how their approach relates to those figures. Furthermore, each example model provides a new figure that is a combination of the architecture of the presented model and the sequence of interactions between the models. This new figure provides more insight into the coupling process and makes the example more transparent for the reader.

Prior to submission we attempted a lessons learned and strengths and weaknesses table with each of the four examples, which did not work well given similar outcomes from each project. We attempted this again, with similar results and therefore leave the lessons learned section within the Discussion. However, as noted in our response to Reviewer 1 each lesson will have an example from presented model accompanying it.

REVIEWER COMMENT

b. Section 2.1 and 2.2 headings sound like case studies and therefore this should be the sub-section title related to case studies. Since you are comparing 4 approaches, the main title of section 2.1, 2.2, 2.3 and 2.4 should illustrate the approach you are addressing in each section as you have done in section 2.3 and 2.4. c.

RESPONSE

Agreed. We swapped different approaches to these headings several times among model names, purposes of the models (as submitted), and others. We have amended the manuscript to describe the example as an approach AND the scientific purpose. This makes the heading lengthy, but much more informative.

While within the manuscript we illustrate different coupling approaches, we do not perform a rigorous comparison of different coupling approaches. This comparison would be best done with different instantiations of the same model(s) using the same data for it to be effective. Instead we seek to demonstrate that with coupling we can answer research questions that cannot be answered without coupling AND that these approaches couple specific process models rather than general models. These specific models provide a level of transparency and depth that are not typically found in more general model integration efforts and therefore facilitate and enable new perspectives and questions to be generated in the science of coupled human-natural systems. Each of the examples illustrates the consequences of modeling feedbacks between human and natural systems at different spatial and temporal scales using different coupling architectures, frequency of communication, and level of coordination. We have clarified

these scalar and architectural differences in the manuscript, but we do not seek to out-line a lengthy literature review and meta-analysis of the different possible approaches to coupling models. In summary, we appreciate this point and have used it to improve the focus and clarity of the manuscript by explicitly describing the coupling architecture for each example including how the models interact with each other, data, and how they are scheduled.

REVIEWER COMMENT

The cases provided in section 2.1 and 2.2 are new models or approaches. As a review paper, I find this somewhat problematic as the idea, I would think, is to appraise pre-existing approaches rather than adding two new approaches. If there are pre-existing approaches that are consistent with those presented in section 2.1 and 2.2, then these should be reviewed. If the authors feel that the approaches they are presenting here are novel and address some of the shortcomings of the approaches in section 2.3 and 2.4, then by all means present your approaches as novel frameworks. However, if they are not, then this article does not strike me as the appropriate platform to present new work (as it is now structured).

RESPONSE

Thank you for drawing attention to some differences in the case study between sections 2.1 and 2.2 with sections 2.3 and 2.4. We face a trade-off in wanting to demonstrate some of the results to the reader to illustrate what types of results are acquired and how they may differ from results not acquired from a coupling (and different types of coupling) between human and natural systems. These results would enable the reader to think about her or his own results and how they may differ under a coupled model. The display of results also provides tangible material for illustrating the lessons learned from the coupling experiences of the group of coauthors. To better align the examples we have edited the example sections to focus explicitly on 1) Model definition and description and 2) Feedback implementation. We have moved the results to supplementary material. In the Feedback implementation section, of each example, we link back to new conceptual Figure 1 and Figure 2 and explicitly outline the architecture of the example.

Neither of the models presented in Section 2.1 or Section 2.2 are new and both have been published (as illustrated by the provided citations). However, all case studies presented illustrate a new approach to modeling human and natural systems. To accommodate the reviewers suggestion we have placed the model case study results in the supplementary material to give the reader further information about the types of results acquired from each model and the types of research questions they can be used to answer.

REVIEWER COMMENT

2. The article states that there is a lack of suitable frameworks to guide the building of coupled models of land systems. Given the wide-ranging expertise of the coauthors here, I would expect the article to present a new framework to tackle these issues based an assessment of the strengths and weaknesses of existing approaches. I understand that you are addressing systems that operate across multiple scales. Nonetheless I would like to see the authors make an attempt to outline a framework that can be applied across scales or at the minimum a list of best practices and core knowledge gaps that are required to be filled. A call to action, if you will. These should be explicitly outlined as they will form the main novel findings of the manuscript.

RESPONSE

We would like to thank the reviewer for pushing us on this issue of providing a framework based on the wide-range of expertise among the coauthors. We hope that the conceptual figures outlining the coupling architecture (Figure 1) and the frequency of communication and level of coordination (Figure 2) address a portion of this request. The coauthors did not find it suitable and thought it was premature at this point in our experiences in model coupling to put forward a generalized framework for others to

follow, but we did come to consensus on proposing a way forward that would help expedite model coupling initiatives and ensure that the products of those initiatives are more interoperable and usable by the science community. We added a section to the Discussion title A Way Forward that contains the following text.

3.4 A Way Forward

[revised manuscript text omitted]

REVIEWER COMMENT

3. The writing style is very loose with statements made that are imprecise, insufficiently explained or qualified. For example, the motivation for coupling on page 5 line 16 – page 6 line 7. Page 6 line 16-17. The coupling approach enables a greater degree of transparency and accuracy in coupled models. There are examples like this throughout the manuscript that need to be made more precise, properly qualified and backed up by citations where appropriate. In addition, many of the sentences are too long and contain multiple arguments. Better to split these into shorter sentences for clarity.

RESPONSE

Thank you for pointing out these areas of improvement. We have altered the text in these sections to be more precise in some cases and qualified in others. We have also qualified other similar statements in the paper. An example to the Reviewer's point on Page 6 line 16-17 is that we changed the text from

"The coupling approach enables a greater degree of transparency and accuracy in coupled models."

To

"Because of the greater degree of openness enabled by these technologies and their modular nature, coupled models enable a greater degree of transparency in how we represent human-natural system models. Whether their relative process richness enables a greater degree of model accuracy remains to be tested."

Furthermore, throughout the manuscript, sentences are shortened.

REVIEWER COMMENT

4. The motivation of the study (stated page 6 line 17 – page 7 line 5) is unclear to me. These are the aims are written as:

Aim 1: We present multiple approaches to coupling land-change and natural-system models to evaluate how alternative approaches to representing feedbacks add value to scientific inquiry into global change and thus generate new insights into the sustainable management of human-environment interactions.

Aim 2: Based on the current state of the science, we categorize conceptual approaches to coupling land-change models with natural-system models that differ across a range of spatial extents and coupling methods.

Aim 3: Using four case studies, we critically assess the influence of land-change processes on natural-system processes and vice versa, focusing on the implications of these feedbacks for system dynamics, the research questions that model coupling enables, and the strengths and weaknesses of the coupling approach.

Aim 4: we describe the lessons learned from the various approaches, the different types of consistency that should be maintained between coupled models as well as the feedbacks represented between the human and natural systems.

Aim 1: I thought the focus was on the land system and land system models, not global change, which is a much broader topic. Aim 2: Okay, a useful aim but I don't think the

paper achieves this. You should make a clearer distinction of the differences between the 4 approaches presented. See comment 1 and 2. Aim 3: I thought the specific aim of the paper was to investigate how human-environment interactions in the land system are captured using coupled modelling approaches.

As it is written, it comes across as quite a verbose research aim. I recommend focusing solely on how human-environment interactions in the land system are captured using coupled modelling approaches. Aim 4: See comment 1 and 2.

RESPONSE

Agreed, we became over excited in what we would achieve with this manuscript. We refined the aims to better match the content of the manuscript and convey this focus at the end of the introduction using the following text:

"We present multiple approaches to coupling land-change and natural-system models and reflect on how their representations of feedbacks add value to scientific inquiry into the dynamics of coupled human-natural systems. We highlight four example models that explicitly represent feedbacks between land-change and natural systems, but vary in their scale of application and coupling architecture. We then present the lessons learned from the modelling research teams, discuss the challenges of representing feedbacks, and then outline a way forward to expedite model coupling initiatives and their subsequent scientific advances."

REVIEWER COMMENT

Specific comments Page 4: Avoid the use of footnotes

RESPONSE

We agree with the reviewer and typically avoid the use of footnotes in all our publications. We have included a single footnote in the submitted manuscript because we believe that the incorporation of the definitions within the text is a distraction from the narrative of the article. The purpose of the footnote is to provide definition and clar-
ification for those who are not used to the nomenclature used in the manuscript and specifically the differences between Earth, ecosystem, and land surface models. If it is acceptable to the journal we would appreciate the inclusion of this content as a footnote rather than integrating the definitions into the text.

Page 6 Line 20: State of the science is incorrect English

Agreed. Thank you for pointing out this error. We have deleted the identified text and edited the paragraph within which the referred text resides to clarify the intentions of the manuscript.

REVIEWER COMMENT

Page 28 Line 18-19. Please qualify this statement.

The statement referred to is as follows:

"These examples provide a level of transparency and detail in the represented processes that is not typically found in larger Integrated Assessment Models (IAMs)."

RESPONSE

Our teams' collective experiential work with and knowledge of IAMs demonstrates that the specific equations and parameters for biophysical processes are often not exposed to the model user, calibrated for specific study areas, or coupled in a way that creates a direct impact-response feedback. Instead, the biophysical processes are either 1) disconnected and act as an independent measurement routine, 2) linked as is often done in climate modelling via prescribed changes in land use and land cover change, or 3) have a simplistic representation that can be useful at very large spatial extents (e.g., globally) but do not match observations well locally.

We have changed the text in this paragraph to replace the text in question with the following:

The focus on specialist-developed models offers a flexible and open approach to answering new questions about feedbacks in coupled human-natural systems, and also facilitates the identification of new types of data required to calibrate and validate the interactions and feedbacks between the two systems. Additionally, in contrast to the monolithic approach to modelling like that taken in integrated assessment models (IAMs), coupled modelling presents an opportunity for increased transparency and detail in the represented processes through more explicit identification and documentation of component interactions and processes.

REVIEWER COMMENT

Page 38 and 39. The conclusion section on interdisciplinary collaboration is valuable and insightful.

RESPONSE

Thank you for this comment and more importantly thank you for your time and effort. Again, your contribution and review is appreciated and has made the manuscript stronger and hopefully more useful for future readers.

---

## Author Response (AR2)

RESPONSE. We would like to thank the editor and three reviewers for their time and effort in reviewing our revised manuscript. Given that Referee 1 (Report 1) and Referee 2 (Report 3) have accepted the manuscript as is, we respond only to Referee #3 (Report 2) below.

I am not an expert in land-use and land-cover change modelling, but do have some experience in modelling of human-environment / social-ecological modelling. I read the current version of the manuscript and the reviews of referees #1 and #2 incl. the responses of the authors.

In the context of modeling feedbacks between human and natural systems the authors discuss coupling approaches of four models where land-change models have been coupled to natural-system models. From these four modelling cases the authors draw general lessons learned and ways forward for the future of coupled Earth system modelling.

I find the overall scope of the paper valuable. I especially like the conceptualization of coupling approaches and feedback types presented in Figure 1 and 2.

RESPONSE. Great. Thanks. We hope the readers of ESD share your opinion.

However, I suggest to revise the paper to clarify and strengthen its arguments.

Overall, I suggest to clarify terms used in the paper. For one thing, these are the terms 'Input', 'Output', 'Data' and 'Coupler'. I refer to Figure 1 and the model description figures (Fig. 3-7).

RESPONSE. Modifications have been made in the use of the terms input and output in relation to Figure 1. We agree this could be slightly confusing for non-modellers reading the paper and we appreciate you pointing this out. We have generalized the text to use data in the majority of cases with reference to model input data and model output at appropriate locations. We have also clarified the use and terms of coupler and coupling framework using the following text:

In model coupling, couplers refer to independent software designed to manage the interaction between two or more models in terms of the passing of data, manipulation of parameters, and scheduling of processes between models and in some cases directing models to data or preprocessing data for use by a model. Typically, couplers are designed for independent research projects using known and available software (e.g., R) and programming languages (e.g., Java, C++, Fortran). When a coupler has been designed for general use across multiple projects, the result is a coupling framework that enables the instantiation of multiple model-coupling projects by others. Like existing modelling frameworks, a coupling framework can speed up the coupling process and facilitate the interaction, adoption, and comparison of different instantiated and coupled models.

This text also captures the reviewers concern about defining a framework for the reader. Again, thanks for pointing these issues out. It certainly helps to clarify these terms for the reader.

By the way, there is no Figure 4 in the paper.

RESPONSE. We appreciate this catch. Thank you. We have revised the document Figure references.

Do you mean with 'Data', externally observed data forcing a model? But how can such 'Data' be updated (c.f. Fig 3) If not, how then 'Data' is different from model 'Input' and 'Output'. And how is 'Data' different from a 'Coupler'? What is a 'Coupler'? And what is the difference between a normal and an integrated assessment coupler? And how can you couple a coupled model without a coupler (c.f. Fig 5). These questions may seem very basic, but I feel clarity in those terms would make the article much more accessible to a wider audience.

RESPONSE. We are slightly unsure how to address this issue, but we have tried to modify the text to be more explicit about the actions of a coupler and clarified and generalized the use of data versus input and output data. Our confusion rests in the fact that data are static unless acted upon by a process (e.g., a model). So while we could differentiate in these diagrams data that provides initial conditions for the model, different data that are used (i.e., input) and processed by different models, and specific data that are generated by models (i.e., outputs). The inclusion of these specifics would add another layer of complexity that would distract from the focus of the paper on model coupling. Furthermore, getting into those details would then suggest that we account for model calibration and validation data with subsequent tangents into sensitivity, uncertainty, and robustness analyses. Instead we offered to provide conceptual types of data that are used and illustrate whether those data remain prescribed (e.g., the climate data in Figure 3 that you mention) versus those data that are manipulated by the various models (e.g., landscape data in Figure 3). We believe that the text accompanying the models (e.g., in the case of Figure 3 in Section 2.1.2 that describes the feedback implementation explicitly informs the reader how the data are manipulated or updated as you request. For example,

" The ABM determines the subsistence requirements for all households in the agent population and passes access to these data as a delimited text string back to the coupler (Figure 3(1)). The coupler then retrieves climate and landscape information and passes data file location information for subsistence requirements to the Agro-Pastoral Yields CA submodel (Figure 3(4))." (Line 278 in the original submission)

We also think that describing in the text how you can couple a coupled model with or without a coupler is somewhat pedantic. The process of coupling two models using a coupler or not is conceptually the same if a third model is added to the process. I hope we have addressed the key issues you point out since overall your assistance has been incredibly helpful.

Another set of terms that deserves more clarity is 'model', simulation environment', 'computational laboratory' and 'framework'. Initially I thought that four models are about to be discussed. But the first "model" is a "computational laboratory" (l.232). The second is a framework (l. 301). This is surely no major issue, however I suggest to clarify these terms, especially "framework", since the article continues with a discussion about another kind of framework.

RESPONSE. As per your suggestion we have defined the term framework in the context of distinguishing the difference between a coupler and coupling framework. We agree simulation environment could be defined, but instead have deleted the term and clarified the text accordingly. We debated the modification or removal of the term computational laboratory in the text. However, Bankes et al. 2002, provides a critical paper in the study and modelling of complex systems that argues that a suite of models should be constructed rather than an integronster (to use the playful term from Voinov and Shugart in Env. Modelling and Software, vol 39, 149-158, 2013). However, the suite of models would not be a duplication of code but rather variations and a systematic increase in complexity and detail that can be turned on and off. Such a computational laboratory is essential to quantify the effects of feedback between two models. Therefore, we make the citation to Bankes et al. (2002) more explicitly associated with the computational modelling term and maintain use of the term in the paper. Consensus among the authors is to not get into the definition of a model or modelling and leave the reader to look at introductory papers and texts on the subject. Thanks for pointing out these terms and pushing us to clarify them.

Here, the article needs more clarity with respect to the framework of coupled models. In the abstract I read: "However, a common framework and set of guidelines to model human-natural systems feedbacks are lacking." (l. 41) Further below, the text says "Novel integrative modelling methods are being developed to create technical frameworks for, and intersecting applications between, these two communities (e.g., Hill et al. 2004, Lemmen et al. 2017, Peckham et al. 2013, Robinson et al. 2013, Collins et al. 2015, Barton et al. 2016)" (l.101) Either these two sentences are contradictory, or two kinds of frameworks are being discussed. Either way, I suggest to clarify. A third alternative could be that the authors feel the lack of a COMMON framework. If this is the case, I would like to read how their proposed framework differs from the ones mentioned and how it is better to eventually become the COMMON framework. Moreover, is the framework that is being envisioned in Sec. "A way forward" a third kind of framework, or how does it relate to the frameworks mentioned so far?

RESPONSE. We appreciate you pointing this out because we are referring to two types of frameworks in these two references. In the abstract we are referring to the lack of a conceptual framework which we provide both through Figures 1 and 2 along with their accompanying text **and** the final section "A way forward". These differ from your point about technical frameworks that aid the instantiation of coupled models. We clarify in the abstract that we are referring to a conceptual framework.

"New coordinating frameworks for next generation coupled modeling of human and Earth systems are being developed within a number of relevant organizations" (l.756)
Scanning recent article of the respective special issue in ESD, Donges et al. submitted "Earth system modelling with complex dynamic human societies: the copan:CORE World-Earth modeling framework" which states to be a "open source software library that provides a framework for developing, composing and running World-Earth models, i.e., models of social-ecological co-evolution up to planetary scales". I wonder and invited the authors to discuss whether and how they would consider such an attempt within their line of desired modelling framework.

RESPONSE. Thank you for pointing out this article. It is quite interesting, relevant, and provides a framework for others to use to create coupled models of human-natural systems. We have included reference to the paper along with those you mentioned above.

Concerning the lessons learned, from my point of view, some of them are quite specific to coupled human-natural systems modelling (lesson 4), others apply to modelling in general (lesson 1). While I absolutely agree that also general modelling lessons are of great importance I suggest to discuss and perhaps order the lessons according to how specific they apply to coupled human-natural models.

RESPONSE. While all of the lessons are provided in the context of model coupling with specific reference to experiences from the four presented example models, it could be argued that Lesson 1 and 2 are the most general and then perhaps Lesson 5, then Lesson 7. However, the coauthors agree that no clear gradient exists among these lessons and the subsequent section of the paper extends from Lesson 8, which therefore cannot be moved. We've done some slight reordering but have agreement among coauthors that declaring a purpose of the ordering to the reader will only initiate argument about the ordering since an underlying gradient was not used to identify the presented lessons. However, we hope that others will extend what we have provided here and further improve upon the experiential knowledge presented in this manuscript

Original Order
*Lesson 1. Leverage the Power of Sensitivity Analysis with Models.*
*Lesson 2. Modelling is an Iterative Process.*
*Lesson 3. Ensure Consistency.*
*Lesson 4. Reconcile Spatio-temporal Mismatch.*
*Lesson 5. Create a Common Language.*
*Lesson 6. Construct Homogeneous Units.*
*Lesson 7. Make Code Open-Access.*
*Lesson 8. Incorporating Feedback Increases Non-Linearity and Variability.*

New Order
*Lesson 1. Leverage the Power of Sensitivity Analysis with Models.*
*Lesson 2. Modelling is an Iterative Process.*
*Lesson 3. Create a Common Language.*
*Lesson 4. Make Code Open-Access.*
*Lesson 5. Ensure Consistency.*
*Lesson 6. Reconcile Spatio-temporal Mismatch.*
*Lesson 7. Construct Homogeneous Units.*
*Lesson 8. Incorporating Feedback Increases Non-Linearity and Variability.*

In the new grouping we also ensured that those lessons that are most related to the Section 3.4 A Way Forward are grouped together (new ordering Lessons 3,4,5, and 6.

Also, the paragraph for lesson 7 needs an argument, why the observed and described problems could be solved with open-access code.

RESPONSE. Understood. The following text has been added to the original Lesson 7:

Moving forward, critical equations, like mass balance equations, and model variables should be made open through coding to provide multiple points for interfacing with other models (specifically human systems models).

L. 209: "The examples are situated at different points along the three dimensions of configuration, frequency of communication, and coordination."
I suggest to illustrate these points, where the examples sit, directly into Figure 1 and 2. In doing so the reader would gain a better initial overview of the examples and how they relate to the given context (framework?).

RESPONSE. Thank you for pointing out this opportunity to improve on the flow
of the manuscript and aid the reader. The text has been modified to include
a one-sentence overview of each example and their relationship to conceptual
Figure 1 and Figure 2 as follows:

The four examples are situated at different points along the three
dimensions of configuration, frequency of communication, and coordination.
The first example uses a coupler in its architecture (similar to Figure 1d,
1f) and achieves two-way coupling (Figure 2) to investigate the effects of
land management on Erosion. Our second example investigates the effects of
land-management on carbon storage using a loose coupling approach with two
models, whereby one acts as a scheduler for the other (Figure 1c) and both
interact with common data to achieve two-way feedback (Figure 2). The third
example uses a coupler to bring together multiple models that share data
(Figure 1d) and create two-way feedback (Figure 2) to investigate changes in
land use and food consumption under climate perturbations. Our forth and
final example uses a coupler-based architecture (Figure 1d) to tightly
couple multiple models to investigate how changes in land use and the energy
system affect terrestrial and atmospheric carbon storage and flux. While all
four examples achieve two-way feedback (Figure 2), most examples originated
with one-way feedback (Figure 2) or were constructed to enable an
investigation of how the incorporation of feedback could alter model
outputs. Collectively, the four examples illustrate how groups of
researchers have attempted to overcome the lack of suitable frameworks for
coupling human and natural systems and the lessons learned for future
representations of feedbacks among human and natural systems.

"One-sided approaches are prone to generating biased results" (l. 67) I encourage the authors to discuss the amount of natural vs. human system components within the presented four cases. Are the selected four cases all in good balance or do (at least) some of them tend to be one-sided. I suggest to use color in the model figures to indicate human, natural and technical model components (couplers, etc.). This would take up this discussion form the introduction which I missed in the rest of the manuscript.

RESPONSE. The reviewer makes a good suggestion and it is one that we have
considered in this manuscript (i.e., better quantifying both the amount of
representation in the human and natural system as well as the effects of
coupling the two systems at varying levels from one-way to two-way on model
outputs). After discussion among our coauthors, we've come to a consensus
that the inclusion of colours in the figure would misrepresent the level of
detail and complexity behind each of the models. For example, in our second
example there is one model representing the human decision-making and land-
management activities but the outcomes of these activities are played out
through four natural system models. In this example the human systems model
is much more complex than each of the natural systems models; however, the
focus of the paper is on the approaches to coupling and not an evaluation or
quantification of the level of detail represented in each coupled model or

the two systems. It is known through our large group of coauthors and their broader networks with which they interact that many models trying to couple human and natural systems are biased and unbalanced in their representation of the two systems, which may be appropriate or inappropriate depending on the circumstance and context of model use. As we mention in the paper, often the bias and weighting is on the natural system with little representation of the human system and we are hoping that our manuscript will help inform those seeking a more balanced modelling approach (or to answer research questions that require a more balanced modelling approach) about how others have tried to couple human and natural systems to make scientific advances.

L.114: "The potential gains from greater coupling are threefold."
Please indicate the three gains more clearly. For example, "the use of many of the Earth's resources by humans alters the state and trajectory of the Earth system" sounds rather as an argument why one should couple models, than a gain from coupling.

RESPONSE. Agreed. Thank you for pointing out this issue of clarity. We have clarified the text as follows:

First, the use of many of the Earth's resources by humans alters the state and trajectory of the Earth system (Zalasiewicz et al. 2015, Waters et al. 2016, Bai et al. 2015). Therefore representing and quantifying the impact of humans on the natural system can determine their magnitude relative to processes endogenous to the natural system as well as provide insight into how to mitigate those impacts through changes in human behaviour.

Lastly, I still agree with referee #2 that the manuscript could benefit from shorting sentences.

RESPONSE. We have edited the manuscript and shortened multiple sentences throughout. Sometimes we just can't help ourselves because we love science! And we appreciate your help to ensure our research is robust and well
10   communicated.

[revised manuscript text omitted]